# Drought Stress during Anthesis Alters Grain Protein Composition and Improves Bread Quality in Field-Grown Iranian and German Wheat Genotypes

**Azin Rekowski** [1]**, Monika A. Wimmer** [1]**, Sirous Tahmasebi** [2]**, Markus Dier** [1]**, Sarah Kalmbach** [1]**, Bernd Hitzmann** [3] **and Christian Zörb** [1,*]

1 Institute of Crop Science, Quality of Plant Products (340e), University of Hohenheim, 70593 Stuttgart, Germany; azin.ghabelrahmat@uni-hohenheim.de (A.R.); m.wimmer@uni-hohenheim.de (M.A.W.); markus.dier@uni-hohenheim.de (M.D.); SarahKalmbach@gmx.de (S.K.)

2 Fars Agricultural and Natural Resources Research and Education Center, Agricultural Research, Department of Seed and Plant Improvement, Education and Extension Organization AREEO, Shiraz 71558-63511, Iran; s.tahmasbi@areeo.ac.ir

3 Institute of Food Science and Biotechnology, Process Analytics and Cereal Science (150i), University of Hohenheim, 70593 Stuttgart, Germany; Bernd.Hitzmann@uni-hohenheim.de

* Correspondence: christian.zoerb@uni-hohenheim.de

**Abstract:** Drought stress is playing an increasingly important role in crop production due to climate change. To investigate the effects of drought stress on protein quantity and quality of wheat, two Iranian (Alvand, Mihan) and four German (Impression, Discus, Rumor, Hybery) winter wheat genotypes, representing different quality classes and grain protein levels, were grown under field conditions in Eqlid (Iran) during the 2018–2019 growing season. Drought stress was initiated by interrupting field irrigation during the anthesis phase at two different stress levels. Drought stress at anthesis did not significantly change total grain protein concentration in any of the wheat genotypes. Similarly, concentrations of grain storage protein sub-fractions of albumin/globulin, gliadin and glutenin were unaltered in five of the six genotypes. However, analysis of protein sub-fractions by SDS polyacrylamide gel electrophoresis revealed a consistent significant increase in ω-gliadins with increasing drought stress. Higher levels of HMW glutenins and a reduction in LMW-C glutenins were observed exclusively under severe drought stress in German genotypes. The drought-induced compositional change correlated positively with the specific bread volume, and was mainly associated with an increase in ω-gliadins and with a slight increase in HMW glutenins. Despite the generally lower HMW glutenin concentrations of the Iranian genotypes and no effect of drought on the concentration of HMW sub-fraction, there was still high specific bread volume under drought. It is suggested that for the development of new wheat cultivars adapted to these challenging climatic conditions, the protein composition should be considered in addition to the yield and grain protein concentration.

**Keywords:** winter wheat; protein composition; bread quality; drought stress; Iranian and German wheat genotypes

## 1. Introduction

Wheat (*Triticum aestivum* L.) is one of the most important staple foods. Due to its unique baking ability, it is cultivated in almost every country of the world, including Germany and Iran [1]. The baking quality is largely determined by the wheat proteins, which can be divided based on their solubility into the fractions albumins, globulins, gliadins and glutenins [2]. Among these fractions, gliadins and globulins together form the adhesive protein 'gluten', which is decisive for the baking ability, while albumins and globulins consist mostly of structural proteins as well as enzymes and hardly affect baking quality [3,4]. The monomeric gliadins are originally divided into the three sub-fractions:

α/β-gliadins, ω-gliadins and γ-gliadins. While gliadins are mainly monomeric proteins that lack cysteine residues (ω-gliadins) or that form intramolecular disulfide bonds (α- and γ-gliadins), glutenins are polymeric proteins. They can be divided into the low molecular weight (LMW) and the high molecular weight (HMW) glutenin subunits [5], which are held together by intermolecular disulfide bonds. According to Jackson et al. [6], LMW can be subdivided into LMW-B, LMW-C (both high in sulfur) and LMW-D (low in sulfur). In many countries, wheat cultivars are classified according to their total grain protein concentration. However, this value does not necessarily correlate with high baking quality, which seems to also depend on the composition of the proteins [3].

Wheat quality parameters such as grain protein concentration, rheological properties and bread making properties are strongly influenced by the genotype and by environmental factors. The presence of some allelic subunits of the HMW glutenin fraction, which is genetically determined, is correlated with the bread making quality [3]. However, gene expression is also highly regulated by environmental factors such as the availability of nutrients at different stages of development, agricultural locations, soil quality, availability of soil water, temperature and other climatic conditions. As a result, the composition of the storage proteins and the subsequent quality aspects can be influenced by both genetic background and environmental conditions [7,8].

Drought stress leads to a limitation of nutrient uptake and transport together with a shortage of water, and has a strong influence on the growth, yield and quality of wheat [9]. In winter wheat, the flowering and grain filling phases are particularly sensitive to drought, which usually shortens the grain filling time and reduces the grain filling rate leading to a reduction in grain yield [3,10–12]. The protein/starch ratio shifts in the direction of protein, resulting in smaller wheat grains with a higher protein and lower starch content [13]. Furthermore, drought stress seems to specifically affect the speed and duration of the accumulation of glutenins and gliadins, as well as the composition of these two fractions [14,15]. Collectively, available evidence thus indicates that drought stress is likely to also affect the baking quality of wheat.

While the increasing occurrence of drought spells, especially during the late season, challenges wheat production in many countries of the world including Germany, the situation is more extreme in Iran, where wheat production in the central and western parts of the country can only be maintained with artificial irrigation [16]. Based on predicted scenarios of climate change, such cultivation conditions could well represent future scenarios of wheat production in parts of central Europe [17]. In view of these challenges, the identification of traits leading to stable and high-yielding wheat genotypes with acceptable quality is becoming increasingly important. Genotypes developed in dry climates may be helpful to identify such traits.

Field studies addressing drought stress are difficult to conduct in Germany, since rainfall (e.g., heavy short rainfall in the form of thunderstorms) is likely to occur especially late in the season during grain filling. Most drought studies are therefore conducted using rain-out shelters, with the drawback of a somewhat reduced light input. In addition, even if rainfall is prevented from reaching the soil, the soil water content is often still relatively high in deeper soil layers, resulting in rather moderate drought conditions under rain-out shelters. In the present study we therefore conducted a field study in the dry area of Iran, where rain is completely absent during the final stage of wheat development (see Figure A1), and where plants thus encounter severe soil and air drought conditions without the need of using rain-out shelters. Four German and two Iranian winter wheat cultivars from two quality classes and with different drought tolerances were used, and the effect of distinct drought stress levels on protein composition and baking quality was assessed.

The present study is based on the following hypotheses: (i) Drought stress alters the composition of grain protein fractions and results in altered baking quality; (ii) German wheat cultivars (even if considered drought tolerant under German climate conditions) are less drought tolerant than Iranian cultivars under conditions of severe drought during grain filling; (iii) Iranian wheat cultivars have been developed under dry climatic conditions, but

they have been selected for making "flat bread" rather than "European-style" bread (based on loaf volume) and may thus express a different protein composition, and especially a lower content of HMW glutenins and bread loaf volume.

## 2. Materials and Methods

### 2.1. Genotype Characteristics

Bread wheat is classified into four (Germany) and two (Iran) quality classes, based on parameters such as volume, dough elasticity, falling number, protein content and sedimentation value. The Iranian cultivars are evaluated under drought stress conditions; quality classes 1 and 2 are roughly comparable to the German classes A and B based on their total protein concentration. Two Iranian (Alvand, Mihan) and four German (Rumor, Impression, Discus, Hybery) winter wheat (*Triticum aestivum* L.) genotypes were used for this study, representing genotypes with higher (class A: Impression, Discus; class 1: Alvand) and lower (class B: Rumor, Hybery; class 2: Mihan) grain protein concentration. Based on previous experiments in a rain-out shelter ([18] for the German cultivars) both class A/and class 1 genotypes are considered more drought sensitive compared to the class B/class 2 genotypes (Table 1).

**Table 1.** Quality classes of the genotypes used and corresponding protein concentration and drought response.

| Genotype | Quality Level | Company | Protein Concentr. | Drought Resistance |
|---|---|---|---|---|
| | | Iranian genotype | | |
| Alvand | class 1 | National Seed and | high | sensitive |
| Mihan | class 2 | Plant Improvement Institute (SPII) | low | tolerant |
| | | German genotype | | |
| Impression | A | I.G. Pflanzenzucht | high | sensitive |
| Discus | A | DSV | high | sensitive |
| Rumor | B | Saaten-Union | low | tolerant |
| Hybery | B | Saaten-Union | low | tolerant |

### 2.2. Trial Location, Experimental Design and Growing Conditions

The field study was conducted at the Research Station of the Seed and Plant Improvement Division in Fars Agricultural and Natural Resources Research and Education Center (AREEO, Darab, Iran). The research field was located near the city of Eqlid, in the province of Fars in Iran (30°53′43.8″ N 52°24′07.5″ E). The field trial was carried out from October 2018 to August 2019.

The soil of the field is a loamy soil consisting of 22% clay, 40% loam and 38% sand. Before sowing, 200 kg ha$^{-1}$ superphosphate (48% $P_2O_5$) was applied. An N fertilization (urea—46% N) of 300 kg ha$^{-1}$ was split into two applications with 150 kg ha$^{-1}$ each at BBCH 24 (tillering) and BBCH 32 (stem elongation) [19]. To control weeds, the herbicide 2,4 dichlorophenoxyacetic acid was applied at the end of the tillering phase. Genotypes were sown with a seed rate of 400 seeds per m$^2$ at a depth of 4 cm.

The total precipitation during the growing season was 515 mm (Figure A1), and occurred mainly in November 2018 and between January and April 2019. Zero precipitation occurred during the late season from June to August 2019. The field trial was a split-split plot design consisting of three irrigation treatments in the main plot and six wheat cultivars in the sub-plot in four replications. Three irrigation treatments were established: well watered (WW) plants were irrigated 10 times between October 2018 and July 2019. Before the anthesis all treatments were well watered. During the anthesis, two times irrigation was applied in well-watered treatment. In moderate stress (MD), plants were irrigated only once, while in severe stress (SD) there was no irrigation. When anthesis was fully completed, all treatments were well watered. Six genotypes × three stress levels × 4 biological replicates (*n* = 4) resulted in a total of 72 treatment plots. The individual plots were laid out with a size of 0.6 × 2.5 m$^2$.

### 2.3. Yield and Total Protein Concentration

At harvest, plants were separated into grains and straw. Grain samples were dried and milled with a tube mill (MM301, Retsch, Haan, Germany). The DUMAS combustion method was used to determine the total N content in the individual flour samples with the aid of the macro elemental analyzer Vario MACRO cube CHNS (Elementar Analysensysteme GmbH, Langenselbold, Germany). The N content was then multiplied by a factor of 5.7 in order to determine the protein concentration.

### 2.4. Extraction of Cereal Proteins

Protein fractions were extracted according to the method of Osborne [20], modified by Wieser and Seilmeier [21]. Since albumins/globulins are not relevant for the baking quality [22], this fraction was extracted, but excluded from further analysis, and only gliadins and glutenins are presented and discussed in this paper. For the extraction, 100 mg of the whole flour was first mixed with 1 mL of extraction buffer 1 (0.067 M $HKNaPO_4$, 0.4 M NaCl, pH 7.6) in an overhead shaker (Multi Bio RS-24 from bioSan, Riga, Latvia) at 20 °C for 5 min, incubated on ice for 10 min with repeated vigorous vortexing, and centrifuged (13,800× $g$, 6 °C, 10 min). The extraction step was repeated two more times, and supernatants containing albumins and globulins were discarded. The remaining pellet was then extracted three times with 0.5 mL of 70% ($v/v$) ethanol at 20 °C for 5 min in an overhead shaker and centrifuged again at 13,800× $g$ (6 °C, 10 min) to yield the gliadin fraction. After a washing step with 1 mL of $dH_2O$ and centrifugation (13,800× $g$, 6 °C, 5 min), the glutenin fraction was extracted three times using 0.8 mL of extraction buffer 2 [2 M urea, 1% ($w/v$) dithiothereitol, 50% ($v/v$) 2-propanol, 0.05 M Tris, pH 7.5] for 5 min in an overhead shaker at 20 °C, followed by incubation at 60 °C for 10 min, cooling to room temperature, and centrifugation at 13,800× $g$ (6 °C, 10 min). All samples were frozen at −20 °C for later use. Two independent extractions were conducted for each sample.

### 2.5. Quantitative Analysis of the Protein Fractions

The concentration of the proteins from the extracted fractions was determined spectrophotometrically (Specord® 50 Plus, Analytik Jena, Jena, Germany) according to the Bradford method by using wheat grain gliadin and glutenin as a reference for calibration [23]. All samples were measured in duplicate to reduce technical errors.

### 2.6. Qualitative Analysis of the Protein Fractions by SDS-PAGE

The Sodium Dodecyl Sulfate Polyacrylamide Gel Electrophoresis (SDS-PAGE) was performed with a standard Dual Cooled Vertical Unit (Hoefer SE 600, Taufkirchen, Germany) according to Laemmli [24], using 12% ($w/v$) polyacrylamide separation gels with a size of 16 × 18 cm and a thickness of 1.3 mm, and a running buffer consisting of 0.192 M glycine, 0.025 M Tris, and 0.01% SDS. A protein amount of 20 μg was loaded into individual lanes. For each gel, a molecular weight marker from 10 to 150 kD was included (Sigma, Taufkirchen, Germany). Running conditions were 400 V for 90 min followed by 480 V for 120 min. The system was cooled to 18 °C using a cooling unit. After the run, SDS-PAGE gels were fixed in 40% ($v/v$) ethanol, 10% ($v/v$) acetic acid, stained in a heated 0.025% Coomassie staining solution (Phast Gel Blue R-350, GE Healthcare, Braunschweig, Germany), washed in $dH_2O$, and destained in 10% ($v/v$) acetic acid solution until the gel background was almost completely decolorized. SDS-PAGE gels were scanned by an image scanner (HP ScanJet 4890, Böblingen, Germany; 300 dpi and 16 bits per pixel, as TIF format).

### 2.7. Evaluation of SDS Gels

The scanned SDS-PAGE gels were evaluated with the Gel Analyzer 2010a program (http://www.gelanalyzer.com/, accessed on 15 December 2020). The gels were analyzed for their color in the "dark on light" mode. For each lane, all existing bands were detected automatically, corrected manually, and numbered according to their respective $R_f$ values

(distance from starting point). For each lane, the sum of the raw volumes (based on pixel intensity) of all bands was set as 100%, and relative intensities of each individual band were calculated and used for comparison between treatments, in order to equalize possible gel to gel staining differences. Statistical calculations were based on four biological replications and two technical replicates.

### 2.8. Flour and Baking Test

Water absorption and dough development time (Farinograph-AT, 810152, Barbender Technologie GmbH & Co. KG; Duisburg, Deutschland) of flour was measured prior to baking experiments. The baking test was performed according to Kieffer et al. [25] with some modifications. To 160 g of flour 1.6 g of dry yeast, 2.4 g NaCl, 1.6 g sugar, 1.6 g fat and 0.016 g ascorbic acid were added. The quantity of water added was based on the water absorption value obtained by the farinograph.

Final amounts of flour were corrected in the way that all flours were adjusted to 14% moisture content for reasons of comparability. All ingredients were added to the kneading chamber, mixed (1 min), kneaded for the determined dough development time (3–5 min) and divided into four portions. The resulting dough was proofed (first proof) at 30 °C and relative humidity of 80% for 25–27 min depending on the dough development time. The dough was then shaped by hand into a ball shape and placed in rings (D: 7.5 cm, H: 4.5 cm). The relaxed dough was again proofed (final proof) at 30 °C and relative humidity of 80% for 55 min plus 15 min scheduled for the forming of the dough. The dough was baked for 25 min at 210 °C in a PICCOLO 1–4 STIR oven (Wachtel GmbH, Hilden, Germany). After one-hour cooling at room temperature, specific volume (VolScan Profiler 600, Stable Micro Systems Ltd., Surrey, UK), hardness (Texture Profile Analyse, WINOPAL Forschungsbedarf GmbH, Elze, Germany), and average pore area (HP ScanJet 5590, Böblingen, Germany) were determined. Freshness retention of samples was estimated after 7 days of storage under $CO_2$ atmosphere by calculating the difference between the hardness of the fresh (1 h after baking) and stored (7 days after baking) bread. Elasticity of the samples was calculated from the ratio of springiness (the tendency of a bread to return to its original shape after it has been compressed) of bread after 7 days of storage to fresh bread (1 h after baking), as determined by a Texture Profile Analyzer (WINOPAL Forschungsbedarf GmbH, Elze, Germany). Baking loss was determined by the mass difference before and after baking. For each sample four technical replicates were used.

### 2.9. Statistical Methods

The data were statistically analyzed with the statistical software R Studio 3.5.0 with a two-factor model. The various test factors were tested for outliers using a box plot. Requirements of the established model including linearity, homogeneity of variance and normal distribution were considered. A two-factor analysis of variance (ANOVA) was used to determine statistical significance between the treatments and followed by multiple contrast tests (Tukey's least significant test at $p < 0.05$ level). A correlation-based principal component analysis was carried out in order to determine positive or negative relationships between the test features.

## 3. Results

### 3.1. Yield and Total Protein Concentration of the Grain

Under well-watered conditions, grain yield ranged from 51.5–65.0 dt/ha (Figure 1). The two Iranian cultivars had the lowest grain yield (Figure 1). Compared to the well-watered controls, grain yield was reduced only marginally by moderate drought stress (0–6.6%), but more strongly by severe drought, where the largest decline was observed for the genotypes Mihan (−24.0%) and Impression (−24.6%), followed by Hybery (−16.6%), Discus (−13.8%), Rumor (−13.4%) and Alvand (−8.3%). However, due to relatively high variations in the field, these effects were not statistically significant. Even under conditions of severe drought, the three highest yields were observed for German genotypes.

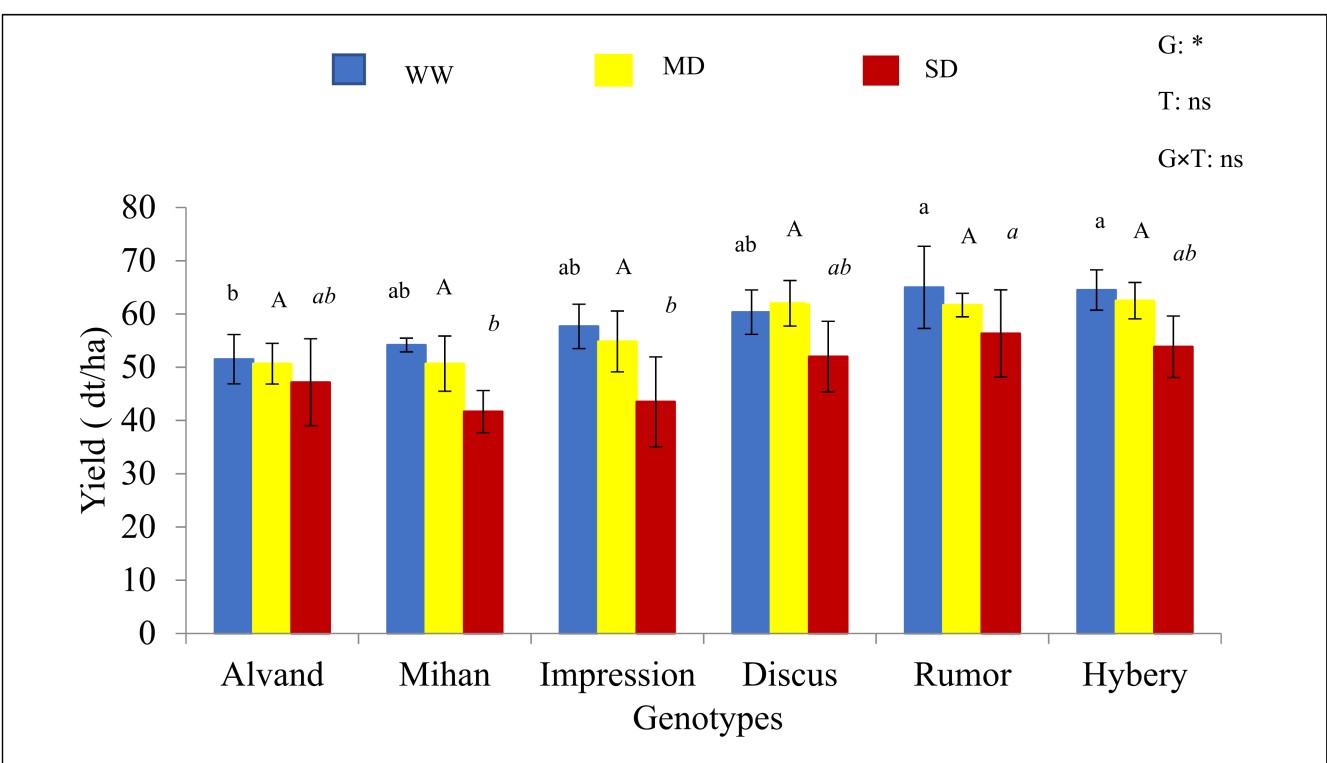

**Figure 1.** Grain yield (dt/ha) depending on the treatments (well watered: WW, moderate drought stress: MD, severe drought stress: SD) of the Iranian (Alvand, Mihan) and German wheat genotypes (Impression, Discus, Rumor and Hybery). The error bars represent the standard errors of the measured values. Different letters within a treatment represent significant differences (small letters: within WW treatment; capital letters: within MD treatment; italic letters: within SD treatment). $p < 0.05$; $n = 4$. Two-way ANOVA results are shown in the upper right corner of the diagram. G: genotype; T: different water treatment; GxT: Interaction between genotype and different water treatment; ns: not significant; *: significant effect.

Under well-watered conditions, total grain protein concentrations ranged between 12.8% (Mihan) and 16.3% (Discus), and only Mihan had significantly lower protein concentrations compared to all other cultivars. No significant difference was observed between the German cultivars in any of the treatments.

Drought stress did not significantly affect total grain protein in any of the six cultivars. However, in four out of six cultivars a slight increase in the protein concentration was observed at the severe stress level (Mihan, Impression, Discus, Rumor). As a result, protein concentrations of Discus and Rumor were significantly higher than those of the other cultivars under severe drought (Figure 2). Only in Hybery was the total protein concentration reduced by drought stress, while it was not affected in Alvand.

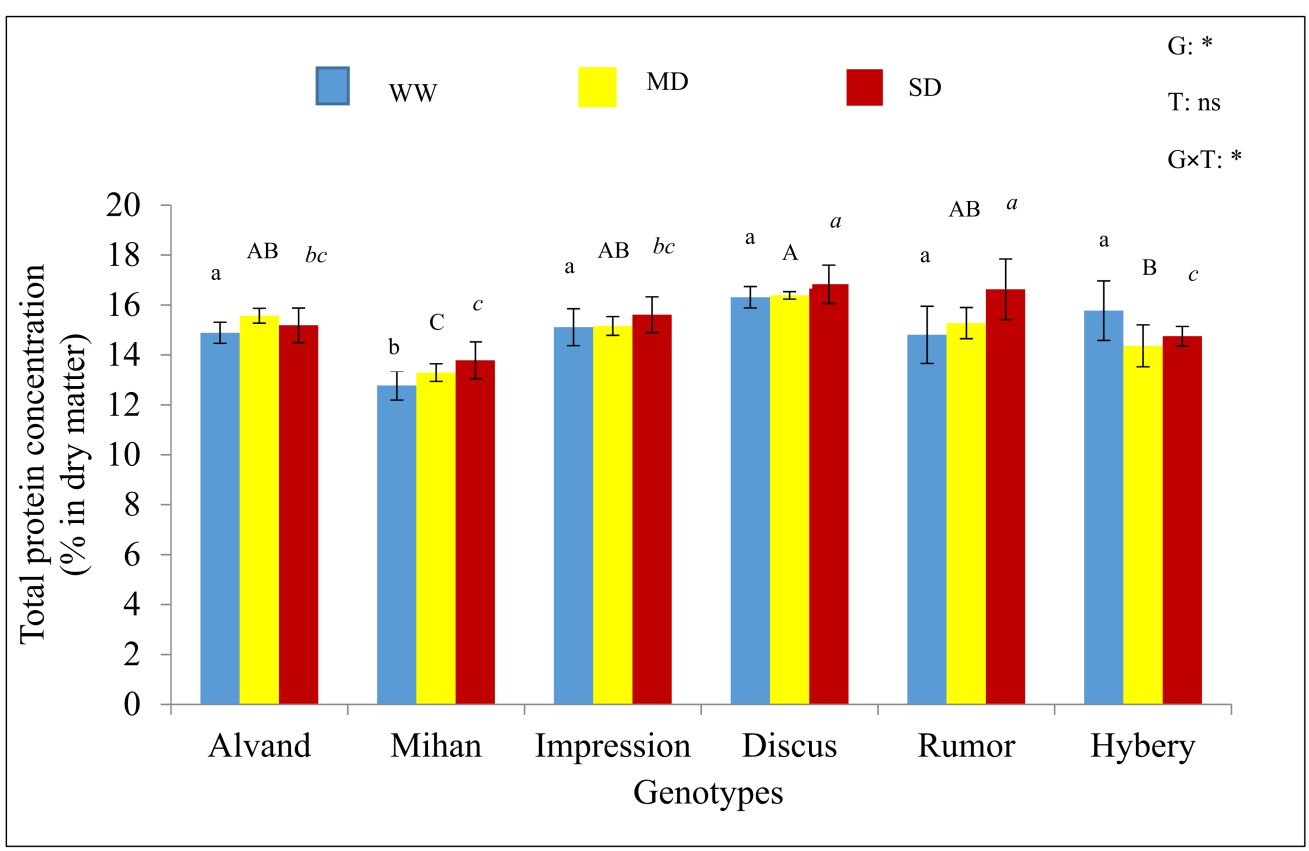

**Figure 2.** Mean values of the total protein concentration (% in dry matter) depending on the treatments (well watered: WW, moderate drought stress: MD, severe drought stress: SD) of the Iranian (Alvand, Mihan) and German wheat genotypes (Impression, Discus, Rumor and Hybery). The error bars represent the standard errors of the measured values. Different letters within a treatment represent significant differences (small letters: within WW treatment; capital letters: within MD treatment; italic letters: within SD treatment). $p < 0.05$; $n = 4$. Two-way ANOVA results are shown in the upper right corner of the diagram. G: genotype; T: different water treatment; GxT: Interaction between genotype and different water treatment; ns: not significant; *: significant effect.

### 3.2. Concentration of Different Fractions

Under well-watered conditions, gliadin concentrations (mg g$^{-1}$ flour) were higher in Alvand, Discus, and Hybery compared to Mihan, Impression and Rumor (Figure 3). Increasing drought stress resulted in slightly enhanced gliadin concentrations in three cultivars (significant for Rumor, tendency for Mihan and Impression) or no change (Alvand, Discus). Interestingly, an opposite behavior was observed for Hybery, where gliadins were reduced compared to the well-watered plants under both moderate and severe (−8%) drought. However, this decrease was statistically not significant.

Glutenin concentrations (mg g$^{-1}$ flour) were significantly lower in the two Iranian genotypes compared to the four German ones under well-watered conditions (Figure 4). Among German genotypes, Impression was outstanding with a very high glutenin concentration, which was also not affected by drought stress (Figure 4). Severe drought stress increased the glutenin level in Mihan (significant) and Discus (tendency) compared to the WW plants, while it did not change it in the other four cultivars. Interestingly, a moderate drought stress increased glutenins in four out of six cultivars, but slightly decreased it in Hybery (Figure 4).

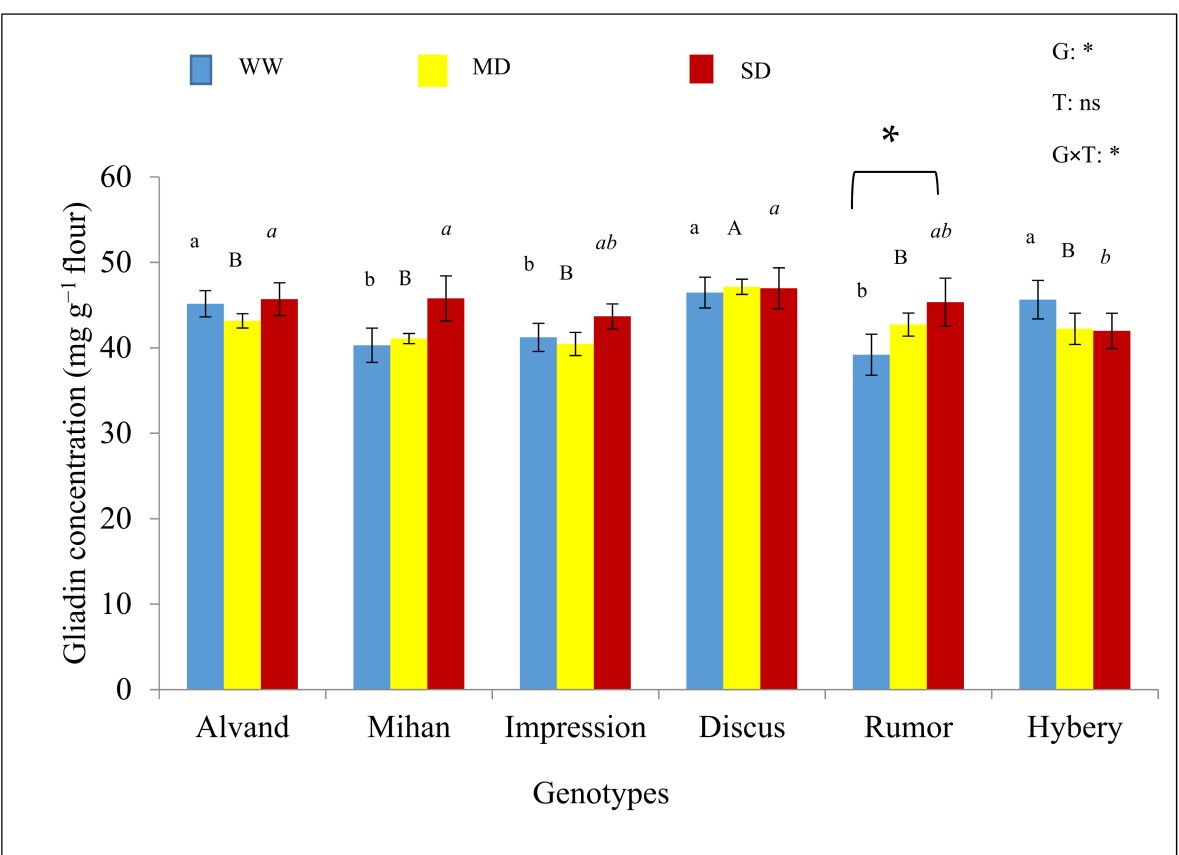

**Figure 3.** Mean values of the gliadin concentration (mg g$^{-1}$ flour) depending on the treatments (well watered: WW, moderate drought stress: MD, severe drought stress: SD) of the Iranian (Alvand, Mihan) and German wheat genotypes (Impression, Discus, Rumor and Hybery). The error bars represent the standard errors of the measured values. Significant differences between treatments within a genotype are indicated by an asterisk (*). The bracket under the asterisk shows which treatments are significantly different from each other. Different letters within the same treatment represent significant differences (small letters: within WW treatment; capital letters: within MD treatment; italic letters: within SD treatment). $p < 0.05$; $n = 4$. Two-way ANOVA results are shown in the upper right corner of the diagram. G: genotype; T: different water treatment; GxT: Interaction between genotype and different water treatment; ns: not significant; *: significant effect.

### 3.3. Composition of the Grain Protein Fractions

Gliadin and glutenin fractions were further analyzed by SDS-PAGE and relative band intensities for each fraction were obtained. In total, between 16 and 20 protein bands were recorded for the gliadin fraction, depending on the genotype. They can be divided into the three sub-fractions ω-gliadins (61.3–45.4 kDa), γ-gliadins (39.3–32.2 kDa) and α/β-gliadins (30.3–20.7 kDa) according to their molecular weight and their position.

The relative share of the gliadin sub-fractions changed with increasing drought stress (Table 2). Under well-watered conditions, the ω-gliadins represented the smallest sub-fraction with 10.7–16.4% of the total gliadins. Their share increased significantly in all six wheat genotypes under drought stress, and in five out of six genotypes this effect was significantly stronger in the severe compared to the moderate drought treatment. The γ-gliadins represented the second largest gliadin sub-fraction under WW conditions, with a share of 23.2–36.1% of the total gliadins. While γ-gliadins were not significantly affected by drought stress in four out of the six wheat genotypes, they were significantly increased under severe drought stress compared to WW plants by 4.9% and 3.3% in Rumor and Discus, respectively. An opposite tendency was observed for α/β-gliadins, which represent the dominant gliadin sub-fraction with a share of 45.7–61.8% of the total gliadins under WW conditions. With the exception of Alvand, α/β-gliadins were reduced under drought stress in all genotypes (significant for Mihan, Discus and Rumor).

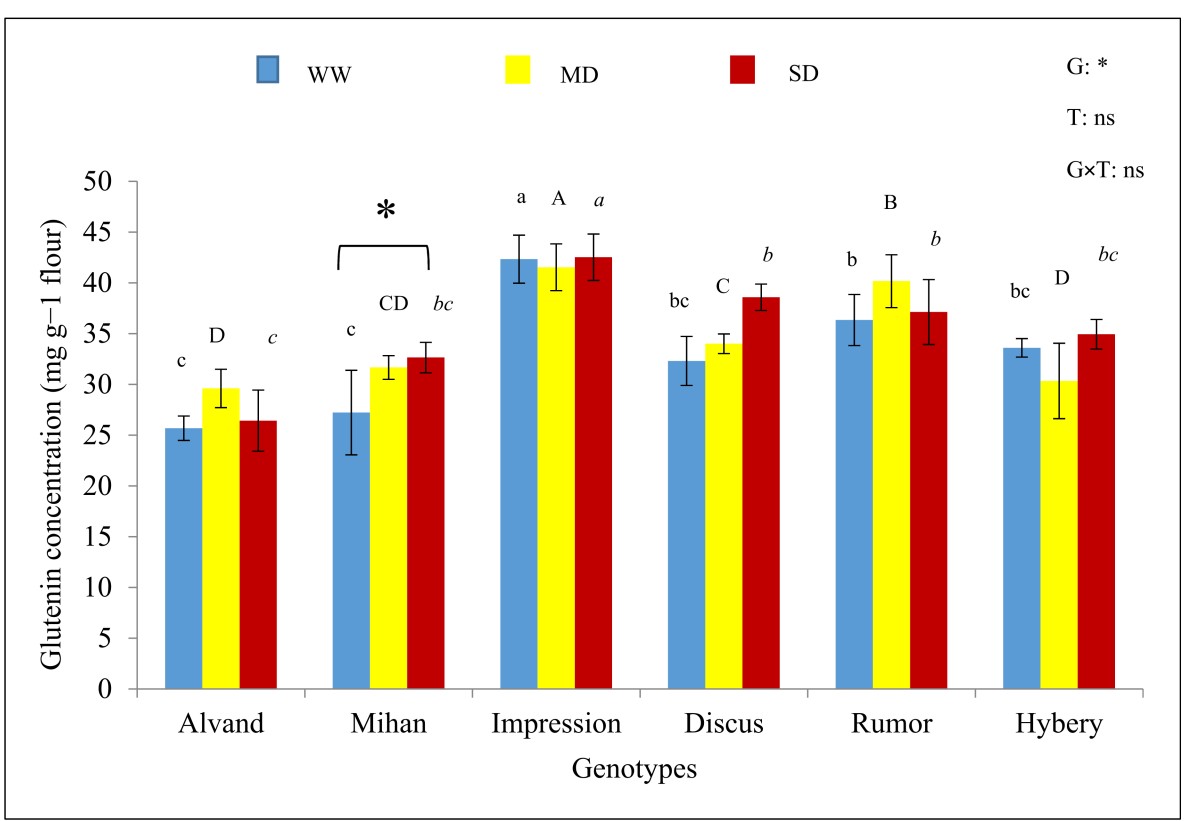

**Figure 4.** Mean values of the glutenin concentration (mg g$^{-1}$ flour) depending on the treatments (well watered: WW, moderate drought stress: MD, severe drought stress: SD) of the Iranian (Alvand, Mihan) and German wheat genotypes (Impression, Discus, Rumor and Hybery). The error bars represent the standard errors of the measured values. Significant differences between treatments within a genotype are indicated by an asterisk (*). The bracket under the asterisk shows which treatments are significantly different from each other. Different letters within the same treatment represent significant differences (small letters: within WW treatment; capital letters: within MD treatment; italic letters: within SD treatment). $p < 0.05$; $n = 4$. Two-way ANOVA results are shown in the upper right corner of the diagram. G: genotype; T: different water treatment; GxT: Interaction between genotype and different water treatment; ns: not significant; *: significant effect.

**Table 2.** Relative share of the gliadin protein sub-fractions (% of total SDS-PAGE lane intensity for each treatment within a variety), depending on the treatments (well watered: WW, moderate drought stress: MD, severe drought stress: SD) of the Iranian (Alvand, Mihan) and German wheat genotypes (Impression, Discus, Rumor and Hybery). For each variety, different letters within the same row represent significant differences between treatments (small letters: within ω-gliadin; capital letters: within γ-gliadin; italic letters: within α/β-gliadin). No letter means no significant differences in this group; $p < 0.05$; $n = 4$.

| Genotype | Alvand | | | Mihan | | | Impression | | |
|---|---|---|---|---|---|---|---|---|---|
| Treatment | WW | MD | SD | WW | MD | SD | WW | MD | SD |
| ω-Gliadin | 14.9 b | 15.5 b | 17.0 a | 12.4 c | 14.6 b | 15.5 a | 16.4 b | 17.2 ab | 17.4 a |
| γ-Gliadin | 36.1 | 37.0 | 48.1 | 34.6 Ab | 31.9 B | 35.5 A | 23.2 | 23.7 | 24.8 |
| α/β-Gliadin | 45.7 | 48.1 | 48.4 | 53.0 *a* | 53.5 *a* | 48.7 *b* | 60.4 | 59.1 | 57.8 |
| Σ | 100 | 100 | 100 | 100 | 100 | 100 | 100 | 100 | 100 |
| Genotype | Discus | | | Rumor | | | Hybery | | |
| Treatments | WW | MD | SD | WW | MD | SD | WW | MD | SD |
| ω-Gliadin | 11.3 c | 12.8 b | 14.4 a | 10.7 c | 12.0 b | 13.1 a | 13.3 c | 14.4 b | 16.5 a |
| γ-Gliadin | 28.2 B | 30.9 Ab | 31.5 A | 29.2 B | 30.9 B | 34.1 A | 24.9 | 25.1 | 23.9 |
| α/β-Gliadin | 60.5 *a* | 56.2 *b* | 54.1 *b* | 60.1 *a* | 57.1 *a* | 52.7 *b* | 61.8 | 60.6 | 59.7 |
| Σ | 100 | 100 | 100 | 100 | 100 | 100 | 100 | 100 | 100 |

The protein bands of the glutenin fraction have been classified into the sub-fractions HMW (128.2–71.5 kDa), LMW-D (63.8–46.0 kDa), LMW-B (39.5–30.0 kDa) and LMW-C (29.7–24.0 kDa) (Table 3). Under WW conditions, HMW glutenins represented 27.3% to 41.3% of the total glutenins, with by far the highest values for the genotype Impression. This protein fraction was not affected by moderate drought, but increased significantly in the four German genotypes under severe drought stress. Interestingly, only a very slight increase was detectable in the Iranian genotypes (Table 3).

The LMW-D glutenins represented the smallest sub-fraction with 1.6% to 6.4% of the total glutenins under non-stressed conditions. This group was absent in the genotype Mihan. The LMW-D glutenins increased significantly under severe drought stress in Discus and Rumor, and slightly in Alvand, while the opposite trend was observed in Impression and Hybery.

The LMW-B glutenins accounted for 26.7% to 39.2% of the total glutenin under control conditions, with highest values for the genotype Mihan, and were overall not affected by drought stress (Table 3) with a slight exception of Rumor at severe stress.

In well-watered plants, 22.8% to 37.9% of the total glutenins belonged to the LMW-C sub-fraction, with by far the highest values for the Iranian cultivar Alvand. In all four German wheat genotypes, the share of LMW-C glutenins was significantly reduced as a result of severe drought stress, while moderate drought had no significant effect (Table 3). A tendency towards lower LMW-C concentrations under severe stress was also observed in the Iranian genotype Alvand, while no effect was seen in Mihan.

Overall, drought-induced changes in glutenin sub-fractions are reflected in the HMW/ LMW ratio, which increased significantly under severe drought stress in all the German, but not in the Iranian genotypes.

**Table 3.** Relative share of the glutenin protein sub-fractions (% of total SDS-PAGE lane intensity for each treatment within a variety), depending on the treatments (well watered: WW, moderate drought stress: MD, severe drought stress: SD) of the Iranian (Alvand, Mihan) and German wheat genotypes (Impression, Discus, Rumor and Hybery). For each variety, different letters within the same row represent significant differences between treatments (small letters: within HMW; capital letters: within LMW-D; bold letters: within LMW-B; italic letters: within LMW-C); No letter means no significant differences in this group; $p < 0.05$. ($n = 4$).

| Genotype | Alvand | | | Mihan | | | Impression | | |
|---|---|---|---|---|---|---|---|---|---|
| Treatments | WW | MD | SD | WW | MD | SD | WW | MD | SD |
| HMW | 27.3 | 27.2 | 28.7 | 33.3 | 35.2 | 35.2 | 41.3 b | 41.4 b | 48.0 a |
| LMW-D | 4.3 | 4.6 | 5.0 | - | - | - | 1.6 | 1.5 | 1.0 |
| LMW-B | 31.2 | 30.2 | 31.0 | 39.2 | 35.6 | 36.0 | 26.7 | 28.0 | 28.1 |
| LMW-C | 37.2 *ab* | 37.9 *a* | 35.3 *b* | 27.5 | 29.2 | 28.8 | 30.6 *a* | 31.9 *a* | 22.8 *b* |
| Σ | 100 | 100 | 100 | 100 | 100 | 100 | 100 | 100 | 100 |
| HMW/LMW | 0.37 | 0.37 | 0.40 | 0.50 | 0.54 | 0.54 | 0.70 b | 0.67 b | 0.92 a |
| Genotype | Discus | | | Rumor | | | Hybery | | |
| Treatments | WW | MD | SD | WW | MD | SD | WW | MD | SD |
| HMW | 27.3 b | 29.2 b | 32.6 a | 32.0 b | 32.4 b | 36.0 a | 31.2 b | 30.9 b | 41.2 a |
| LMW-D | 5.1 B | 5.3 B | 6.8 A | 6.4 B | 6.5 B | 10.6 A | 5.2 | 4.8 | 3.9 |
| LMW-B | 36.3 | 34.5 | 35.2 | 32.8 **ab** | 33.7 **a** | 30.8 **b** | 32.9 | 33.4 | 31.7 |
| LMW-C | 31.4 *a* | 31.0 *a* | 25.4 *b* | 28.5 *a* | 27.8 *a* | 22.7 *b* | 30.7 *a* | 30.8 *a* | 23.2 *b* |
| Σ | 100 | 100 | 100 | 100 | 100 | 100 | 100 | 100 | 100 |
| HMW/LMW | 0.37 b | 0.41 b | 0.48 a | 0.47 b | 0.48 b | 0.56 a | 0.45 b | 0.45 b | 0.70 a |

### 3.4. Baking Quality

In order to determine the influence of drought stress on the final technological properties of the bread, a standard baking test was carried out. Compared to the control, water absorption, elasticity, baking loss and mean pore area did not change significantly under drought (data not shown). In all six wheat genotypes, the specific bread volume increased under drought stress (well watered < moderate < severe drought stress), and this effect was significant at the severe stress level (Alvand, Impression, Rumor, Hybery) or already at the moderate stress level (Mihan, Discus) (Table 4). There was no significant difference in specific volume between German and Iranian genotypes across all treatments. Across all treatments, Impression exhibited a significantly higher specific volume compared to all other genotypes, while it was lowest in Discus and Rumor (Table A1).

With the exception of Alvand, a drought-induced decrease in freshness retention was observed, even though the effect was statistically significant only for Rumor (Table 4). It is important to mention that a lower freshness retention value corresponds to an improved bread quality. Hardness also slightly decreased under drought (significant for Discus).

**Table 4.** Baking quality parameters depending on the treatments (well watered: WW, moderate drought stress: MD, severe drought stress: SD) of the Iranian (Alvand, Mihan) and German wheat genotypes (Impression, Discus, Rumor and Hybery). For each variety, different letters within the same genotype represent significant differences between treatments (small letters: specific volume of the bread; capital letters: freshness retention; italic letters: hardness); No letter means no significant differences in this group; $p < 0.05$; $n = 4$.

| Genotype | Alvand | | | Mihan | | | Impression | | |
|---|---|---|---|---|---|---|---|---|---|
| Treatments | WW | MD | SD | WW | MD | SD | WW | MD | SD |
| Specific volume (mL/g) | 3.3 b | 3.4 b | 3.7 a | 2.9 b | 3.3 a | 3.4 a | 3.5 b | 3.6 b | 3.8 a |
| Freshness retention (N) | 13.1 | 16.7 | 15.6 | 14.8 | 12.7 | 12.4 | 17.1 | 15.3 | 15.4 |
| Hardness (N) | 4.6 | 5.1 | 4.6 | 5.1 | 4.4 | 4.0 | 5.0 | 4.9 | 4.6 |
| Genotype | Discus | | | Rumor | | | Hybery | | |
| Treatments | WW | MD | SD | WW | MD | SD | WW | MD | SD |
| Specific volume (mL/g) | 2.8 c | 3.0 b | 3.3 a | 2.8 b | 2.9 b | 3.0 a | 3.2 b | 3.2 b | 3.4 a |
| Freshness retention (N) | 17.0 | 16.2 | 13.8 | 18.6 A | 13.8 BB | 13.3 B | 15.4 | 14.4 | 14.0 |
| Hardness (N) | 6.0 *a* | 5.0 *ab* | 4.2 *b* | 6.1 | 5.6 | 5.1 | 5.2 | 4.4 | 4.8 |

### 3.5. Principal Component Analysis

In order to determine the relationship between storage protein fractions, their subfractions and the baking quality (Table 4), a correlation-based principal component analysis (PCA) was carried out. The first two principal components together explained 60.1% of the total variance (PC1: 36.9%, PC2: 23.23%) (Figure 5).

The PCA allowed to visualize the separation of the different genotypes (Figure 5), and to identify the traits mainly correlated with the PCs (Figure 6). In the scores plot (Figure 5), PC1 separated the genotypes Alvand, Mihan, Discus and Rumor on the negative side from Hybery and Impression on the positive side. Impression is especially associated with high values of PC1. Within each genotype, the samples from well-watered Discus, Rumor and Mihan plants were bundled furthest to the left, those of moderately stressed plants in the middle, and those of severely stressed plants furthest to the right. For the genotypes Impression, Hybery and Alvand, well-watered and moderately stressed samples were not clearly separated, but severely stressed samples again clustered furthest to the right. PC2 tended to separate the German cultivars on the positive side and the Iranian cultivars on the negative side, with partial overlaps between the genotypes, suggesting that PC2 might be related somehow to the cultivar origin. Comparison of the scores plot and the correlation circle (Figure 6) indicates that PC1 is positively correlated with ω-gliadin concentrations, high specific volume, high HMW concentrations and a high HMW/LMW ratio, and negatively correlated with LMW-B. It seems that PC2 is positively correlated with

α/β-gliadin and negatively with LMW-C. Moreover, there is a highly significant positive correlation between ω-gliadins and the specific bread volume.

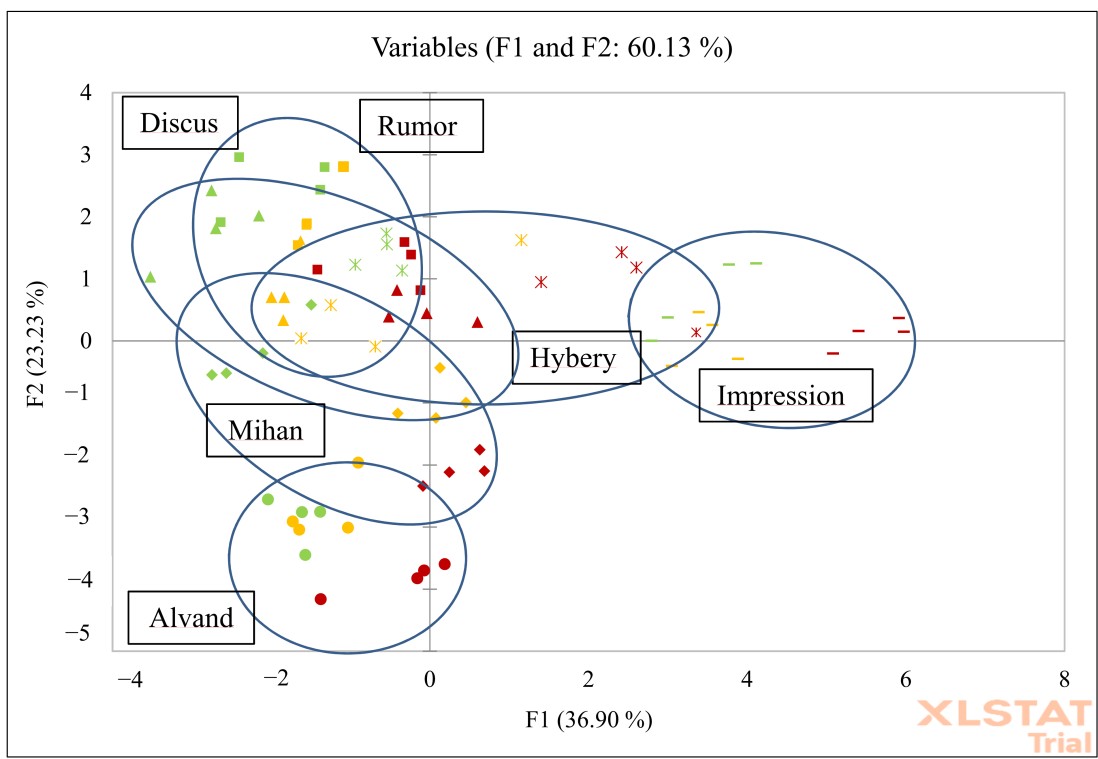

**Figure 5.** Biplot representation of the principal component analysis according to Pearson. Outlined data points are observations of the Iranian (Alvand, Mihan) and German wheat genotypes (Impression, Discus, Rumor and Hybery). Colours indicate the treatments well-watered: green, moderate drought: yellow, severe drought red; symbols indicate the six genotypes (circle: Alvand; diamond: Mihan; triangle: Discus; square: Rumor; star: Hybery; line: Impression).

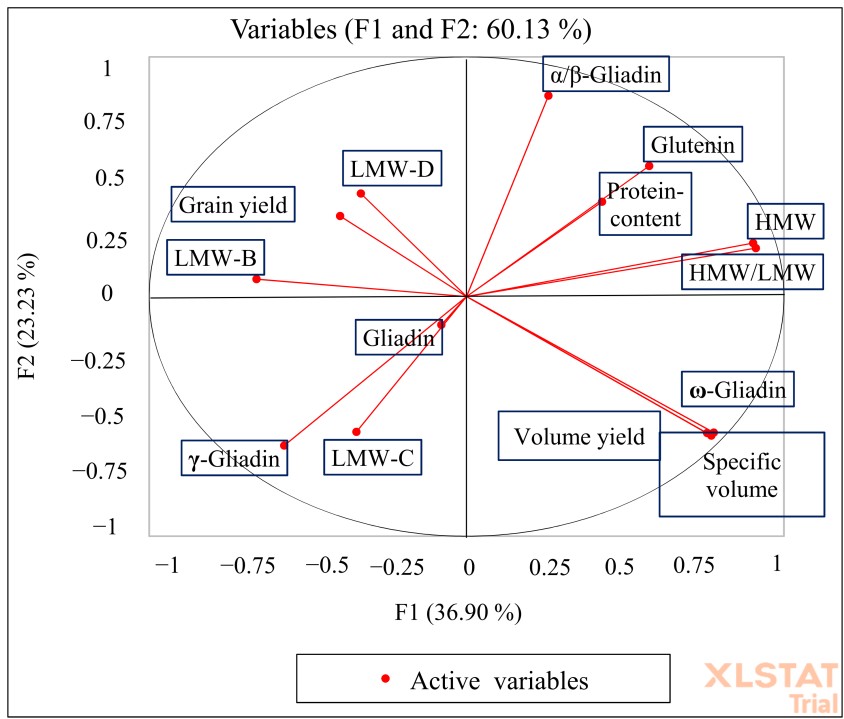

**Figure 6.** Correlation circle examined by Pearson correlation with *p* = 0.05.

## 4. Discussion

### 4.1. Drought-Induced Improved Baking Quality Is Related to Changes in Protein Composition Rather Than Total Protein Content

This study was conducted based on the hypothesis that severe drought during the grain filling phase of wheat increases grain protein content, alters the protein composition and thus affects the bread baking quality.

Classification of wheat into quality classes is largely based on the total grain protein content. Similar to other studies [18,26], no significant effect of drought stress on total grain proteins was detected. Nevertheless, a slight increase in total grain protein content was observed under severe stress in five out of six genotypes, which might be related to the production of fewer, but larger grains under stress, in line with the observed slight increase in thousand grain weight under severe stress, as well as with other studies [15,18]. Larger grains may represent a benefit for successful germination of the next generation [27]. In addition, it was also suggested that a drought-induced enhanced ABA production was negatively correlated with grain starch concentration, and may favour the accumulation of grain storage proteins [28,29].

Despite the lack of a significant effect on total grain protein under severe drought, strong changes in the protein composition and an improved baking quality were observed in all six wheat genotypes. Specifically, increases in the sub-fractions ω-gliadins as well as decreases of α/β-gliadins and LMW-C glutenins were observed across all genotypes, while HMW glutenins increased only in the German genotypes.

It is already known that baking quality is not only determined by the total amount of wheat proteins, but also by the protein composition of the various protein fractions [3,8], and it was proposed that ω-gliadins and HMW-glutenins in particular have a strong influence [21,30]. Both ω-gliadins and HMW glutenins belong to the protein sub-fractions with low sulfur content [21] and therefore have only few disulfide bonds. Their stress-induced increase may result in a reduced degree of polymerization of the gluten protein network. In addition, ω-gliadins also have a high glutamine content (45% to 53%), which is involved in the formation of glue proteins by the formation of temporary hydrogen bonds and hydrophobic interactions [31–33]. In the present study, ω-gliadins were strongly positively correlated with the specific bread volume (Figure 6), and they were the only protein sub-fraction consistently and significantly increased in all six genotypes, while HMW glutenins increased only in the four German genotypes. In addition, the specific bread volume was significantly higher in the two Iranian genotypes compared to the German genotypes Rumor and Discus. This indicates that the ω-gliadin sub-fraction may have been the most relevant protein fraction with respect to baking quality in the present study. This is somewhat surprising since usually glutenins (especially HMW and LMW-C) are considered very important. However, neither of these two glutenin sub-fractions was affected by drought in the Iranian genotypes. On the other hand, total glutenins were less abundant in the Iranian compared to the German genotypes, possibly related to the fact that Iranian genotypes are selected for the production of "flat bread" rather than "loafy bread", and that a high glutenin content is less important for this feature.

Previous studies suggest that an increase in the HMW/LMW glutenin ratio is correlated with a higher baking volume [8]. This effect was confirmed in the present study, but only for the German wheat genotypes under severe drought stress, and the concomitant significant reduction of LMW-C glutenins in the German varieties under drought may have additionally contributed to the shift of this ratio. However, this effect cannot explain the improved specific bread volume observed in the Iranian genotypes. Collectively, these results suggest that severe drought stress during grain filling positively affects baking quality of wheat. The ω-gliadins may be more relevant, and HMW glutenins less relevant for the bread baking quality than previously expected. There is no indication that the slightly higher total grain protein content was decisive for the significant drought-induced improvements of baking quality.

### 4.2. Baking Quality Might Be Somewhat Negatively Related to Drought-Induced Increases in γ-Gliadins and LMW-D Glutenins

In all treatments Discus and Rumor had the highest total protein concentration, but the lowest specific bread volume. These two genotypes were the only ones where γ-gliadins and LMW-D glutenins were significantly increased under severe drought stress. It is possible that the increase in the latter two sub-fractions counteracted the significant increase in ω-gliadins and HMW-glutenins and led to an overall less optimized dough strength and lower specific bread volume. The genotype Impression on the other hand, had the by far highest specific volume of all six wheat genotypes, combined with a medium protein content in all treatments (Table A1). This genotype was outstanding because of its high level of ω-gliadins and HMW-glutenins, and very low levels of γ-gliadins and LMW-D glutenins, which were also not altered under drought. The high baking quality of this genotype can be attributed to a relatively high ω-gliadin content and a high HMW/LMW ratio.

### 4.3. Iranian Genotypes Are Not Generally More Tolerant to Severe Drought Stress Than German Genotypes

This field study was conducted based on the hypothesis that wheat genotypes developed in very dry climates (e.g., Iran) have an intrinsically higher drought tolerance compared to those from moderate climates (e.g., Germany), and may represent candidates for use in central Europe in the future when late season drought spells are likely to increase significantly.

Different wheat genotypes were selected representing drought tolerant and drought sensitive cultivars, based on preceding studies [18]. It is interesting that this classification was only confirmed for the two Iranian genotypes, while drought-induced yield reductions were similar for the four German genotypes, irrespective of their reported drought tolerance or sensitivity. This is relevant, since it indicates that reported tolerance to drought may not be consistently observed when plants are grown in a different environment, where other factors such as soil type or light intensity may play a role. It highlights the need for testing/selection of genotypes under different environmental conditions in order to choose the optimum adapted genotype under changing climatic conditions. With respect to yield stability, only the tolerant Iranian genotype Alvand performed better, while the yield reduction under stress was even stronger in Mihan compared to all German genotypes. The Iranian genotypes originate from a different gene pool than the German ones, and it is known that the composition of wheat proteins depends on the genetic background [34]. However, it seems that neither the genetic background alone, nor the climatic conditions under which a genotype was developed, automatically provide a significant advantage under drought conditions.

### 4.4. Drought Tolerant Iranian Genotypes Have a High Baking Quality and Might Be Suitable Candidates for Drought-Prone Areas in Germany

Unexpectedly, the Iranian genotypes both exhibited a very good bread baking quality, despite the fact that these genotypes were not selected for this trait and overall had a rather low concentration of glutenins. It seems that these genotypes compensate for the low amount of HMW glutenins by high levels of ω- and γ-gliadins, which are strongly increased under severe drought. With respect to baking quality, at least the tolerant genotype Alvand may thus be considered a possible candidate for cultivation in highly drought-prone areas of central Europe. However, its total grain yield was the lowest of all tested genotypes under well-watered and drought conditions, raising the question of whether it would be competitive enough economically for the European market.

### 4.5. Conclusions and Outlook for Further Research

Severe drought stress during the grain filling phase may positively affect bread baking quality in winter wheat, even though it reduces total grain yield. The beneficial effect is highly correlated with increased amounts of ω-gliadins rather than HMW glutenins, but additional protein sub-fractions also seem to have significant effects. Different mechanisms

may be relevant for baking quality in genotypes of different origins, and quality parameters such as protein concentration and quantity of individual fractions may not be sufficient for the selection of wheat genotypes under drought conditions. Among German genotypes, Impression had a high potential in terms of baking quality under drought stress, while genotypes such as Rumor and Discus were rather unsuitable despite their stable protein content. Paying attention to the protein composition may help to develop more efficient wheat varieties with an optimal baking quality when there is a lack of water. However, the results also highlight the necessity to evaluate the drought tolerance of wheat genotypes under their respective environmental conditions in the field.

**Author Contributions:** Conceptualization, C.Z. and B.H.; methodology, A.R.; software, S.K.; validation, M.D., C.Z. and M.A.W.; resources, C.Z.; data curation, A.R.; writing—original draft preparation, A.R. and S.K.; writing—review and editing, C.Z., M.A.W., B.H. and S.T.; supervision, C.Z. and S.T.; project administration, C.Z.; funding acquisition, C.Z. All authors have read and agreed to the published version of the manuscript.

**Funding:** This research was funded by the Foundation fiat panis and Food Security center (FSC), University of Hohenheim, which is supported by the German Academic Exchange Service (DAAD) with Funds from the Federal Ministry of Economic Cooperation and Development (BMZ) of Germany, grant number DAAD 57160040.

**Institutional Review Board Statement:** Not applicable.

**Informed Consent Statement:** Not applicable.

**Data Availability Statement:** Not applicable.

**Acknowledgments:** We thank Parviz Salehi for his skillful technical assistance in carrying out experiments in the field. We thank Herbert Götz for his excellent technical assistance for preparing the baking test. We are also immensely grateful to Martin Zahner for his help in the Dumas method and Christiane Beierle for her non-stop support during lab work. Further thanks to the Research Station of the Seed and Plant Improvement Division in Fars Agricultural and Natural Resources Research and Education Center (AREEO, Darab, Iran) for providing facilities for field experiments. Azin Rekowski gratefully acknowledges the financial support provided by the Foundation Fiat Panis and Food Security Center (FSC), University of Hohenheim, which is supported by the German Academic Exchange Service (DAAD) with Funds from the Federal Ministry of Economic Cooperation and Development (BMZ) of Germany, grant number DAAD 57160040.

**Conflicts of Interest:** The authors declare that the research was conducted in the absence of any commercial or financial relationships that could be construed as a potential conflict of interest.

## Appendix A

**Table A1.** Significant differences within a respective treatment (well watered (WW), moderate stress (MS), severe stress (SS) across the six examined wheat genotypes. Values represent the means of the four biological replicates. Different letters within a column show significant differences between the varieties ($p < 0.05$).

| Genotype | Treatments | | |
|---|---|---|---|
| | **WW** | **MD** | **SD** |
| $\omega$-Gliadin (% of the total gliadin content) | | | |
| Alvand | 14.9 b | 15.5 b | 17.0 ab |
| Mihan | 12.4 d | 14.6 c | 15.5 c |
| Impression | 16.4 a | 17.2 a | 17.4 a |
| Discus | 11.3 e | 12.8 d | 14.4 d |
| Rumor | 10.7 f | 12.0 e | 13.1 e |
| Hybery | 13.3 c | 14.4 c | 16.5 b |

**Table A1.** *Cont.*

| Genotype | Treatments | | |
|---|---|---|---|
| | WW | MD | SD |
| **HMW Glutenin (% of the total glutenin content)** | | | |
| Alvand | 27.3 d | 27.2 e | 28.7 e |
| Mihan | 33.3 b | 35.2 b | 35.2 c |
| Impression | 41.3 a | 41.4 a | 48.0 a |
| Discus | 27.3 d | 29.2 d | 32.6 d |
| Rumor | 32.0 bc | 32.4 c | 36.0 c |
| Hybery | 31.2 c | 30.9 cd | 41.2 b |
| **LMW-C Glutenin (% of the total glutenin content)** | | | |
| Alvand | 37.2 a | 37.9 a | 35.3 a |
| Mihan | 27.5 c | 29.2 cd | 28.8 b |
| Impression | 30.6 b | 31.9 b | 22.8 d |
| Discus | 31.4 b | 31.0 bc | 25.4 c |
| Rumor | 28.5 c | 27.8 d | 22.7 d |
| Hybery | 30.7 b | 30.8 bc | 23.2 d |
| **Specific volume (mL/g)** | | | |
| Alvand | 3.3 b | 3.4 b | 3.7 b |
| Mihan | 2.9 d | 3.3 bc | 3.4 c |
| Impression | 3.5 a | 3.6 a | 3.8 a |
| Discus | 2.8 e | 3.0 d | 3.3 d |
| Rumor | 2.8 e | 2.9 e | 3.0 e |
| Hybery | 3.2 c | 3.2 c | 3.4 c |

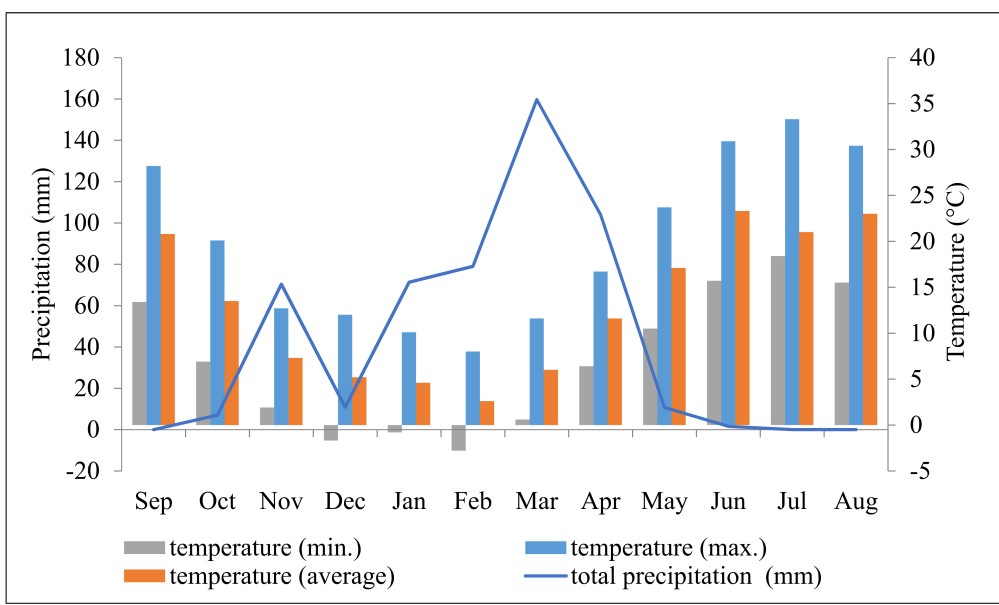

**Figure A1.** Monthly precipitation (bars) and average temperature (lines) during the experimental years for Eghlid. Data source: Eghlid synoptic station, Fars Meteorological Department, I.R. of Iran Meteorological Organization, Eghlid, Iran.

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
