# Peer review of "Drought Stress during Anthesis Alters Grain Protein Composition and Improves Bread Quality in Field-Grown Iranian and German Wheat Genotypes"

_applsci, doi:10.3390/app11219782_

Round 1

Reviewer 1 Report

In the case of increasingly frequent dry periods in Europe, this research seems to be very beneficial, as it can contribute to the fight against the effects of drought. However, I have a few comments:

  1. References - citations in the text and the literature list do not meet the journal's requirements
  2. Figure 3 - where there is an asterisk in the figure with the parenthetical referred to in lines 305-306.

Author Response

We now have changed our manuscript according to the your suggestion. Thank you for helping improving the manuscript.

Reviewer 1:

  1. References were formatted
  2. Figure 3 was corrected (the asterisk is now available in the figure and in the headline

Reviewer 2 Report

The article is interesting and with constant climate change and reduced rainfall, droughts are becoming more and more common, such research is absolutely necessary and desirable. Here are some suggestions:

  1. Line 287: Please edit the caption under figure 2. It should start with a line below.
  2. There are no homogeneous groups listed in the tables. When a factor has no significant influence on a trait, the group "a" should be written everywhere. The lack of a description of the groups is misleading. 
  3. Lines 368-370: You have written that the LMW-B glutenins were overall not affected by drought stress. What about genotype Rumor?
  4. Line 371: The values given in the text are not the same as those in Table 3. In text there is 22.7% and 37.9% and in the table 27.5% and 37.2%. 
  5. Line 394: Reference to Table 4 is missing.
  6. Table 4: There are no homogeneous groups for specific volume for the Impression and Hybery genotypes and for Freshness retention for the Rumor genotype for WW. Please correct the table description - letter W is missing from WW description - well watered for Impression and Hybery genotypes. 

Author Response

Thank you for your help to improve the manuscript. We now have changed our manuscript according to your suggestion

  1. Line 287: Please edit the caption under figure 2. It should start with a line below.

#Done

  1. There are no homogeneous groups listed in the tables. When a factor has no significant influence on a trait, the group "a" should be written everywhere. The lack of a description of the groups is misleading. 

# Done, we included a sentence to clarify this “no letter means no significant differences in this group”

  1. Lines 368-370: You have written that the LMW-B glutenins were overall not affected by drought stress. What about genotype Rumor?

# We have corrected this and mentioned the slight change in Rumor

  1. Line 371: The values given in the text are not the same as those in Table 3. In text there is 22.7% and 37.9% and in the table 27.5% and 37.2%. 

# 37,9 in the table is Alvand (MD) and 22,7 is Rumor (SD) it is inaccordance to the table now

  1. Line 394: Reference to Table 4 is missing.

# corrected

  1. Table 4: There are no homogeneous groups for specific volume for the Impression and Hybery genotypes and for Freshness retention for the Rumor genotype for WW. Please correct the table description - letter W is missing from WW description - well watered for Impression and Hybery genotypes. 

# corrected